# Redox imbalance is related to HIV and pregnancy

**Vanessa Martinez Manfio**[1]*, **Karen Ingrid Tasca**[2], **Jessica Leite Garcia**[3], **Janaina de Oliveira Góis**[1], **Camila Renata Correa**[3], **Lenice do Rosário de Souza**[1]

**1** Department of Tropical Diseases- São Paulo State University–UNESP/Botucatu-Brazil, Botucatu, São Paulo, Brazil, **2** Department of Microbiology and Immunology- São Paulo State University–UNESP/Botucatu-Brazil, Botucatu, São Paulo, Brazil, **3** Department of Medical Clinics- São Paulo State University–UNESP/Botucatu-Brazil, Botucatu, São Paulo, Brazil

* vanessamanfio@hotmail.com

**Data Availability Statement:** All relevant data are within the paper and its Supporting Information files.

**Funding:** This study was funded by Fundação de Amparo à Pesquisa do Estado de São Paulo -

## Abstract

Redox imbalance may compromise the homeostasis of physiological processes indispensable to gestational development in HIV-infected women. The present study aims to evaluate markers of the redox system in the development of pregnancy of these women. HIV-positive pregnant women, HIV-negative pregnant women and non-pregnant were studied. Redox markers superoxide dismutase (SOD), catalase (CAT), protein carbonylation and malondialdehyde (MDA) were assessed at first or second trimester, third trimester and postpartum from pregnant and from non-pregnant women. According to the longitudinal analysis model, CAT activity was increased in the postpartum in HIV-positive women and before delivery in HIV-negative women. Increased carbonylation was observed in the pre-delivery period of HIV-negative pregnant women and MDA concentrations were higher in HIV-positive pregnant women compared to those non-infected by HIV at all times. According to the factorial model, higher SOD and CAT activities were observed in HIV-positive women in the initial months of pregnancy and in non-pregnant women. Carbonylation at third trimester was more evident in HIV-negative pregnant women. MDA levels were higher in HIV-positive pregnant women. Increased oxidative stress may occur in HIV-infected pregnant women. Nevertheless, the HIV virus is not solely responsible for this process; instead, mechanisms inherent to the pregnancy seem to play a role in this imbalance.

## Introduction

Human redox imbalance may be intensified by HIV infection and antiretroviral therapy (ART), which affects patients' clinical, immunological and nutritional status [1–3].

The chronicity of inflammation caused by HIV replication, which occurs even in individuals with viral suppression, overcomes physiological pro-oxidant mechanisms [4]. To counteract the oxidative damage, there is a high consumption of antioxidant molecules, which in part explains the deficiency of antioxidants commonly found in this population [1, 4]. Changes in total antioxidant capacity and reduced levels of antioxidant enzymes (superoxide dismutase

FAPESP and Coordenação de Aperfeiçoamento de
Pessoal de Nível Superior - CAPES.

**Competing interests:** The authors have declared
that no competing interests exist.

[SOD], catalase [CAT] and glutathione), antioxidant vitamins (A, C and E) and micronutrients (zinc, iron and copper) are also observed [1, 5–7].

Despite the improve of clinical, viral and immunological parameters of HIV-infected individuals provided by ART, these drugs affect mitochondrial activity and increases reactive oxygen species (ROS) [8, 9]. Some authors have also described increased serum levels of malondialdehyde (MDA), genotoxic, hydroxiperoxic, and intense oxidized protein damage, as well as glutathione, vitamins, carotenoids, and total antioxidant losses, in HIV-infected individuals, especially after the beginning of ART [4, 10, 11].

During the gestational period, an increase in ROS seems to be a physiological condition, as their levels fluctuate through the different gestational trimesters according to changes in the fetal demand of nutrients and oxygen, and peaks with labor [12]. In a recent study, Basu et al. [13] evaluated placentas of healthy pregnant women and detected an increase in lipid peroxidation products in the first trimester of pregnancy compared to the second and third trimesters, with increased antioxidant activity in these periods. According to the authors, the progressive increase of oxidative stress during pregnancy also leads to an increase in antioxidants compounds in the placenta, resulting in a control of redox system.

Increased oxidative activity is associated with complications such as preeclampsia, gestational diabetes and intrauterine growth restriction [14, 15]. Hernandez et al. [16] evaluated placentas of women undergoing ART (zidovudine) in the first trimester of pregnancy and reported high levels of MDA and carbonylated protein, as well as a reduction in mitochondrial activity and cellular apoptosis [16]. These findings may be the etiological basis of gestational complications and provide data regarding the toxicity of the ART administered to HIV-infected pregnant women.

There are few studies evaluating the ROS and inflammatory profile of HIV-infected pregnant women and their implications for maternal and newborn health. Nevertheless, the importance of oxidative stress analysis during pregnancy is well recognized, as its increase may be related to premature birth [17] and endothelial dysfunction in women with preeclampsia [18, 19], among other obstetric complications [20]. Thus, the aim of the present study was to evaluate the influence of both HIV and pregnancy factors, in the redox system of these women, by using the unprecedented combination of two models of analysis, one factorial and the other longitudinal.

## Methods

### Casuistic and study design

We included 39 women seen at the Specialized Outpatient Service for Infectious Diseases "Domingos Alves Meira" (SAEI-DAM)–Botucatu Medical School Complex (FMB)-UNESP and at the Botucatu Blood Center (HCFMB) São Paulo State, Brazil. The inclusion was carried out according to the universe of eligible pregnant HIV patients who were attended at our Health Service. After acceptance to participate in the study, the patients were allocated in one of the four groups: G1: 13 HIV-positive pregnant women, G2: 10 HIV-negative pregnant women, G3: 10 non-pregnant HIV-negative women, and G4: six non-pregnant HIV-positive women.

Pregnant women were initially enrolled during the first trimester (up until the 13th week) or second trimester (14th to 27th week) of gestation. For the longitudinal segment of the study, sample collection was performed during the first or second trimester (M0), during the third trimester around the 37th week of gestation (M1) and during the puerperium, after three months of delivery (M2). The transversal segment of the study comprised non-pregnant women (control) enrolled during the menacme, from whom samples were collected only once.

Women with cancer, autoimmune diseases, organ transplants, immunosuppressed, or undergoing continuous use of corticoids were excluded.

## Data collection

Sociodemographic data were obtained during an interview and included age at enrollment, use of vitamin supplements, gestational age, duration of HIV infection and use of ART. Data on T CD4+ lymphocyte count and HIV plasma viral load (VL) were obtained from medical records. This study was approved by the Research Ethics Committees of the Botucatu Medical School (FMB), UNESP (n° 3.200.047/2016) and all the participants signed Written Informed Consent.

## Laboratory exams

Peripheral blood (10 mL) was collected using anticoagulant tubes (EDTA) from all participants. The blood was centrifuged at 1500 rpm for 10 minutes at room temperature and the plasma was stored in a -80° C until specific laboratory tests were performed.

**Measurement of oxidative stress products.** Quantification of MDA was performed using 250 μL of plasma and 750 μL of 10% trichloroacetic acid for protein precipitation. The samples were centrifuged at 3000 rpm for 5 minutes and the supernatant collected. Thiobarbituric acid (TBA) 0.67% (1:1) was added to the supernatant and the samples were heated for 15 minutes at 100° C. MDA reacted with TBA at 1:2 MDA-TBA ratio and, after cooling, the read was performed at 535 nm in a Spectra Max 190 microplate reader (Molecular Devices®, Sunnyvale, CA, USA). The concentration of MDA was obtained through the molar extinction coefficient ($1.56x10^5$ M-1 cm-1) and samples' absorbance. The final result was expressed in nmol/g of protein [21].

Carbonylation was measured by a method adapted from Mesquita et al. [22] in 100 μL of plasma to 100 μL of 2,4-dinitrophenylhydrazine (DNPH) (10 mM in 2 M HCl). Samples were incubated for 10 minutes at room temperature and then 50 μL of NaOH (6 M) was added and incubated again for 10 minutes at room temperature. The reading was performed at 450 nm in a microplate reader and the result obtained from the absorbance of the samples and the molar extinction coefficient ($22000$ $M^{-1}$ $cm^{-1}$). The result was expressed in nmol/mg protein.

**Measurement of antioxidant enzymes.** The SOD activity was determined by the technique of Crouch et al. [23], based on the ability of the enzyme to inhibit nitroblue-tetrazolic (NBT) reduction by free radicals generated by hydroxylamine in alkaline medium (pH 10). Hydroxylamine generates $O_2^-$ flow from NBT to blue-formazane at room temperature. When the sample was added, the rate of NBT reduction was inhibited according to the percentage of SOD present in the sample. Enzyme activity was expressed in atomic mass unit (U)/mg total protein.

The CAT activity was determined in phosphate buffer pH 7.0 using 0.5 mL of sample and hydrogen peroxide (30%) according to Aebi H. [24]. Readings were performed at 240 nm.

**Determination of total protein.** The determination of total protein was performed in plasma for correction of antioxidant enzymes and oxidizing products, using the Bioclin® colorimetric commercial kit and its absorbance was measured in a spectrophotometer at 545 nm.

## Statistical analysis

Descriptive analysis of sociodemographic data was performed. Longitudinal analysis for oxidants and antioxidants markers was performed considering only pregnant women (G1 and G2) in the different moments. ANOVA followed by Tukey's multiple comparison test was used for data that presented symmetrical distribution, and Gamma distribution followed by Wald's multiple comparison test was used for asymmetric data.

The factorial analysis was performed by comparing the four groups at each moment. Similarly, two-way ANOVA and interaction followed by Tukey's multiple comparison test were performed for variables with symmetrical distribution. For data with asymmetric distribution, a Gamma distribution model was fitted to the same test, followed by the Wald multiple comparison test. The significance level of 5% and all analyses were performed using SAS for Windows, version 9.4, with the assistance of the FMB-UNESP Research Support Office.

## Results

Sociodemographic, clinical and laboratory data are showed in Table 1. The groups were homogeneous regarding the following variables: gestational and laboratory variables, use of supplements and scheme of ART.

The average age of the HIV + pregnant group was lower than non-pregnant women (both HIV + and HIV-, p = 0.002). Despite this, we do not believe that this difference influences the markers analyzed here, since an average age of 25 years and an others of 31 and 39 years would most likely not lead to profound changes in the antioxidant capacity and formation of pro-oxidants in these women.

**Table 1. Sociodemographic and clinical data of patients included.**

|  | Pregnant HIV+ (G1, n = 13) | Pregnant HIV- (G2, n = 10) | Non-pregnant HIV- (G3, n = 10) | Non-pregnant HIV+ (G4, n = 6) |
|---|---|---|---|---|
| **Age (years); $\acute{x}$±SD** | 25±7,16 [a] | 31±4,33 [ab] | 31±7,60 [b] | 39±9,69 [b] |
| **Gestational variables** |  |  |  |  |
| Age at enrollment ($\acute{x}$, weeks) | 15 | 18 | NA | NA |
| Age at childbirth ($\acute{x}$, weeks) | 38 | 40 | NA | NA |
| **Use of supplements; n (%)** |  |  |  |  |
| Folic Acid | 8(61) | 8(80) | NA | NA |
| Ferrous sulfate | 5(38) | 3(30) | NA | NA |
| Multivitamin | 0 | 3(30) | NA | NA |
| **Laboratory variables** |  |  |  |  |
| T CD4+ gestational $\acute{x}$±SD | 655±294 | NA | NA | 892±101 |
| T CD4+post-partum/ non-pregnant $\acute{x}$±SD | 607±233 | NA | NA | 892±101 |
| Undetectable VL/ pregnancy; n (%) | 6(46) | NA | NA | - |
| Undetectable VL/ post-partum/ non-pregnant; n (%) | 10(76) | NA | NA | 6(100) |
| **HIV diagnosis and ART; $\acute{x}$ (min-max)** |  |  |  |  |
| Time of HIV diagnosis (months) | 68(1–228) [a] | NA | NA | 280(216–384) [b] |
| Time on ART (months) | 47(1–205) [a] | NA | NA | 248(168–336) [b] |
| **Gestational ART; n (%)** |  |  |  |  |
| NRTI +PI | 7 (54) | NA | NA | NA |
| NRTI+NNRTI | 3 (24) | NA | NA | NA |
| Other | 3 (24) | NA | NA | NA |
| **Post-partum/ non-pregnant ART; n (%)** |  |  |  |  |
| NRTI +PI | 3 (23) | NA | NA | 2 (33) |
| NRTI+NNRTI | 6 (46) | NA | NA | 4 (66) |
| No medication | 4 (31) | NA | NA | 0 |

NA = non applicable; ($\acute{x}$ = mean; n = number of individuals; SD = standard deviation; min = minimum value; max = maximum value; VL = HIV viral load; ART = antiretroviral therapy; NRTI = nucleoside reverse transcriptase inhibitor; NNRTI = non-nucleoside reverse transcriptase inhibitor; PI = protease inhibitor; T CD4+ lymphocyte count expressed in cells/mm$^3$. Different letters represent differences between groups (p < .05).

All HIV-infected women, pregnant and non-pregnant, were on ART at the time of inclusion in the study, however four pregnant HIV+ abandoned or paused medication in the postpartum period. Most pregnant women were using a combination of nucleoside reverse transcriptase inhibitor (NRTI) with protease inhibitor (PI)—along with integrase inhibitor for two of them—followed by NRTI combined with non-nucleoside reverse transcriptase inhibitors (NNRTI). During the postpartum, the combination of NNRTI and NRTI was indicated for most women. Although the groups are homogeneous in relation to the ART class used, as some women discovered the diagnosis together with pregnancy, this group had a shorter time of infection (p = 0.009) and, consequently, a shorter time on ART (p = 0.012) in relation to HIV+ non-pregnant women.

Mean CD4+ T lymphocyte count in the gestational and postpartum periods of HIV-infected women were above 350 cells/mm$^3$ and, although viral loads were undetectable for most women, during the postpartum four women presented viremia above the limit of detection (40 RNA copies/mL).

Additionally, none of the included women had co-infections or opportunistic diseases. Regarding the presence of other comorbidities and drugs use, only one patient had diabetes and other two were smokers (data presented in S1 Data), which did not influence our results.

## Results–longitudinal model

To evaluate the redox status, endogenous antioxidants were investigated such as the enzymatic activities of SOD and CAT, and also the oxidants markers, MDA and protein carbonylation. In the G1 group, we detected increased CAT levels in M2 compared to M1 (p = 0.03), while for the G2, samples collected in M0 exhibited the lowest enzymatic activity compared to other moments and to G1 ([M0 x M1 p = 0.0004], [M0 x M2 p = 0.001] and [G1 x G2 p = 0.05] Fig 1A). However, no changes in SOD were observed throughout the progression of pregnancy and postpartum in either group (Fig 1B). Regarding carbonylation, the only difference found was for G2, between M0 and M1 (p = 0.01) (Fig 2A), while MDA was higher at all G1 moments compared to G2 ([M0 p<0.0001, M1 p = 0.0005 and M2 p<0.0001] Fig 2B).

## Results–factorial model

Antioxidant activity is shown in Fig 3. Considering the "pregnancy factor" at M0, HIV infection increased the CAT activity compared to uninfected women (p = 0.02) (Fig 3A). By fixing

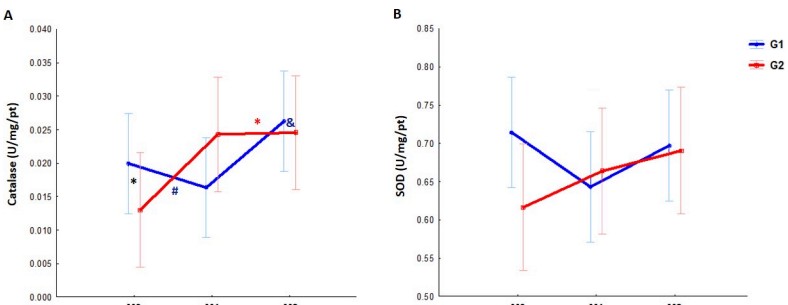

**Fig 1. Longitudinal evaluation of antioxidant enzyme activity (superoxide dismutase [SOD] and catalase [CAT]) from 23 pregnant women infected and not infected by HIV.** Bars: 95% confidence interval. M0: moment zero (first or second trimesters), M1: moment one (third trimester), M2: moment two (post-partum). G1 (red): HIV-positive women, G2 (blue): HIV-negative women. *A;* * *red: difference between M1 and M2 (p = 0.03), # blue: difference between M0 and M1 (p = 0.0004); & blue: difference between M0 and M2 (p = 0.001).* * *black difference between G1 and G2: M0 p = 0.05. Statistical analysis: ANOVA.*

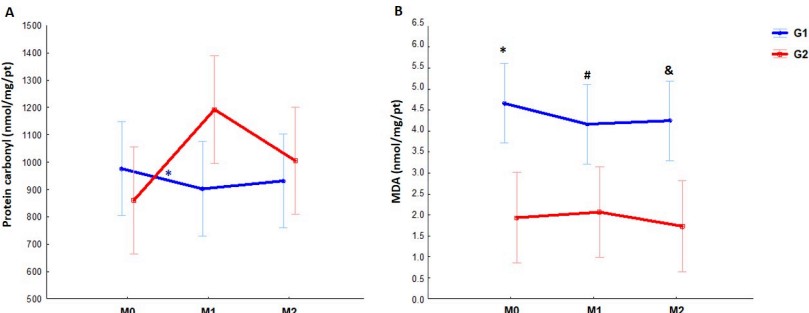

**Fig 2. Longitudinal evaluation of oxidants markers (protein carbonyl and malondihaldeide [MDA]) from 23 pregnant women infected and not infected by HIV.** Bars: 95% confidence interval. M0: moment zero (first or second trimesters), M1: moment one (third trimester), M2: moment two (post-partum). G1 (red): HIV-positive women, G2 (blue): HIV-negative women. *A; * blue: difference between M0 and M1(p = 0.01). B; G1 × G2: * black: M0 p<0.0001, # black: M1 p = 0.0005 and & black M2 p<0.0001. Statistical analysis: Gamma Distribution. followed by Tukey-Kramer post hoc tests (SOD) and Gamma Distribution (CAT).*

the infection, CAT activity was lower for pregnant women compared to non-pregnant at M0 (p = 0.01) and M1 (p = 0.004). At M2, CAT levels were higher for HIV-negative women at postpartum compared to those who were not pregnant and or infected (p = 0.01). Comparing infection and non-infection (control group), according to the non-pregnancy factor, the CAT levels were higher in the infected patients (p = 0.0008).

Regarding SOD at M0, when we fixed the "pregnancy factor", HIV-positive patients had higher consumption of this enzyme (p = 0.04) (Fig 3B). At M1, when "infection factor" was not considered, pregnant women had higher levels of SOD compared to non-pregnant women (p = 0.006). SOD levels were lower during postpartum in uninfected women than in women who were not pregnant (p = 0.006). When "pregnancy factor" was not considered, SOD was higher in HIV-positive than in HIV-negative women (p = 0.007).

Uninfected pregnant women presented higher carbonylation during M1 when compared to infected patients (p = 0.01) (Fig 4A). When the "infection factor" was disregarded,

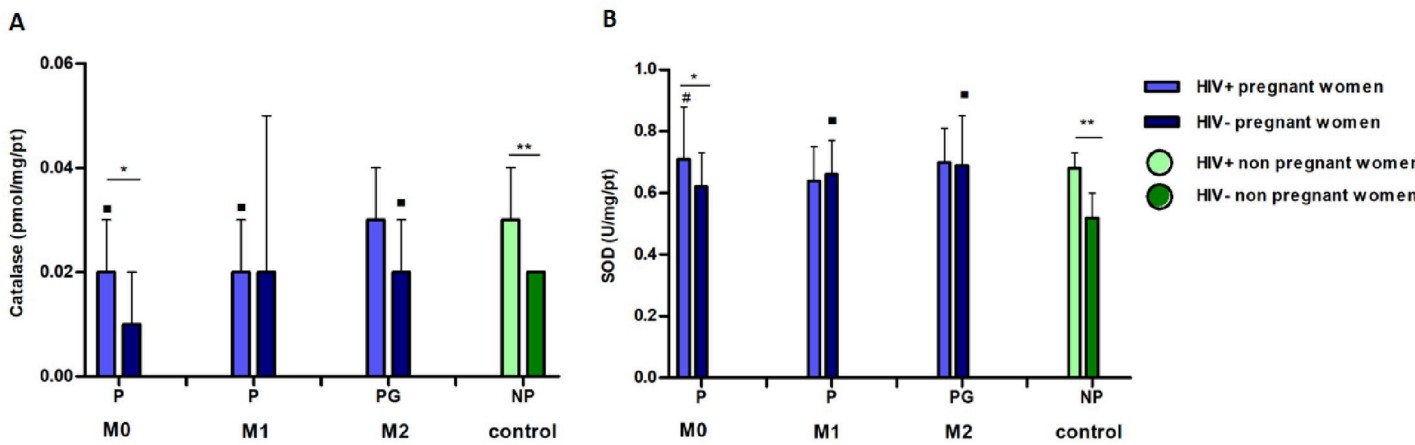

**Fig 3. Factorial evaluation of antioxidant enzyme activity (superoxide dismutase [SOD] and catalase [CAT]) from the 39 included women.** M0: moment zero (first or second trimesters), M1: moment one (third trimester), M2: moment two (post-partum). **M0 and M1** P: pregnancy. **M2** PG: postpartum **Control** NP: non-pregnant. NI: not infected, I: infected. *A:M0 *G = i x ni p = 0.02, - I = g x ng p = 0.01. M1 - I = g x ng p = 0.004. M2 - NI = pg x ng p = 0.01. Control ** NG = I x ni p = 0.002. B:M0 *G = i x ni p = 0.04,M1-NI = g x ng p = 0.006. M2 - NI = pg x ng p = 0.006. Control ** NG = i x ni p = 0.004. Statistical analysis: ANOVA followed by Tukey-Kramer post hoc tests (SOD) and Gamma Distribution (CAT).*

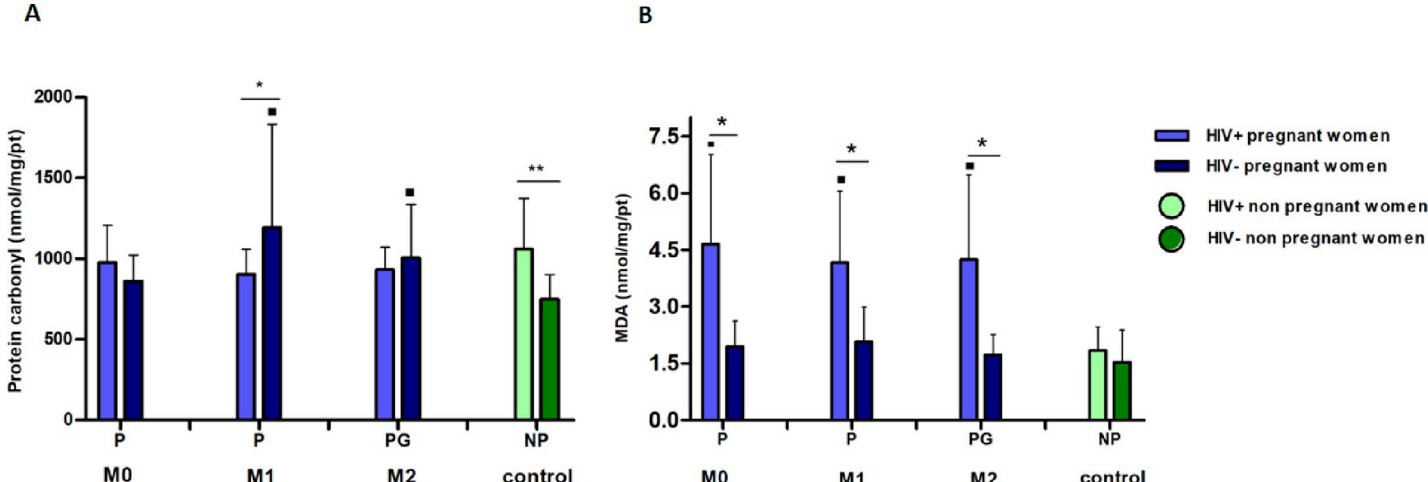

**Fig 4. Factorial evaluation of oxidants markers (protein carbonyl and malondihaldeide [MDA]) from the 39 included women.** M0: moment zero (first or second trimesters), M1: moment 1 (third trimester), M2: moment 2 (post-partum). **M0 and M1** P: pregnancy. **M2** PG: postpartum. **Control** NP: non-pregnant. NI: not infected, I: infected. **A:M1** *G = i x ni p = 0.01, - NI = g x ng p = 0.003. **M2** - NI = pg x ng p = 0.004. **Control** **NG = i x ni p = 0.002.**B**: **M0** *G = i x ni p = 0.002, - NI = g x ng p = 0.001. **M1** *G = i x ni p = 0.003, - NI = g x ng p = 0.009. **M2** *G = i x ni p<0.0001, - NI = g x ng p = 0.003. Statistical analysis: Gamma Distribution.

carbonylated protein levels were higher for pregnant women at M1 (p = 0.0003) and postpartum (p = 0.004).

In the assessment of lipid peroxidation, higher levels of MDA were observed in HIV-infected pregnant women compared to uninfected in all the moments (M0 p = 0.0002, M1 p = 0.003 and M2 p <0.0001) (Fig 4B). When the "infection factor" was fixed, pregnant women had higher MDA levels than non-pregnant women in all the moments analyzed (M0 p = 0.001, M1 p = 0.009 and M2 p = 0.003).

## Discussion

Oxidative stress and the inflammatory process play a crucial role in the progression of AIDS. The variation in the redox balance at different stages of infection is explained by viral effects and also by ART, considering that despite treatment, there is an intense generation of ROS with deregulation of endogenous antioxidant molecules [2, 3, 8, 9, 25, 26]. Pregnancy is also a state of intense oxidative stress [27] which is regulated by various physiological mechanisms in order to maintain the balance between radical production and the antioxidant capacity that will allow the normal progression of gestation and fetal growth [28].

Both in the longitudinal and factorial models, in the first months of pregnancy (M0), women infected with HIV presented higher SOD activity than uninfected ones. However, pre and postpartum (M1 and M2) activity of this enzyme were similar in both groups. Thus, pregnancy by itself does not seem to change this enzymatic activity. In the factorial model, when we excluded the "pregnancy factor", infected patients also showed higher SOD activity than uninfected, which indicates the influence of the virus on this mechanism. Unlike the longitudinal model, when we considered only HIV-negative women, there was greater SOD activity in pregnant women than in non-pregnant women, demonstrating that there may be oscillations in SOD activity throughout the gestational period only in uninfected women. Also in relation to endogenous antioxidant enzymes, in the factorial model, HIV-positive women also presented higher CAT activity compared to HIV-negative patients, and when we considered all HIV-positive patients, the "gestation factor" showed lower activity of this enzyme, whose increase was only observed in the postpartum period in the longitudinal model. Although

pregnant women present lower CAT activity than non-pregnant women, and this activity becomes more intense as the pregnancy progresses in all women.

Considering only HIV infection, the results described herein are corroborated by Ibeh et al. [29], who also showed higher SOD and CAT activity in people living with HIV than in uninfected individuals. On the other hand, Osuji et al. [30] described that HIV-positive individuals presented lower SOD and CAT enzymatic activity than the uninfected group both before the introduction and after 12 months of ART. However, even though these patients did not reach the activity levels observed in uninfected subjects, after six and 12 months of ART there was a significant increase in these enzymes, which may be a consequence of lower production of ROS and free radicals due to the progressive increase of CD4+ T cells and decreased VL after ART [30].

When we only consider pregnancy, the literature is quite controversial. While the CAT activity was decreased in our population of infected pregnant women, SOD activity was increased to protect against oxidative stress. However, in a study with healthy pregnant women [31], there was high activity of both SOD and CAT. In another study [20], the oxidative damage was higher in the second trimester of pregnancy.

These controversial results can be attributed to the characteristics of the studied populations. In the present study, most of participants had a good immune response. Considering that high endogenous antioxidant activity may play a protective role in TCD4+ cells and thus, limit the infection [29], women included in this study may be able to regulate pro-oxidant products generated by pregnancy and/or infection.

Regarding the carbonylation results, pregnant women not infected by HIV showed an increase of this protein damage before delivery, in both analysis, and the pregnancy status seems to increase these levels only in uninfected women. Thus, among uninfected pregnant women, the findings of increased carbonylation, especially in the pre-delivery period, suggest that this intense oxidative imbalance occurs transiently and is restricted to this period.

Li et al. [32] reported higher carbonylation in diabetic pregnant women in the first and third gestational trimesters. Other factors may also influence the redox status, such as anemia, which is very common in pregnant women. Tiwari et al. [33] observed that increased lipid and protein peroxidation may lead to a higher risk of gestational complications, so it is important to understand the oxidative state in these women.

Excluding pregnancy, HIV-positive women had higher levels of this protein damage. Thus, both infection and pregnancy seem to influence this pro-oxidant action. Considering only HIV infection, our results of higher carbonation and decreased SOD and CAT activities in HIV-positive women are corroborated by other studies that show higher oxidative stress in this population compared to uninfected people, which is a consequence of both the viral effect [34] and ART [8, 9, 35].

MDA is the main product used to estimate lipid peroxidation. In the present study, we reported higher MDA levels in HIV-positive pregnant women compared to pregnant HIV-negative women. Considering all HIV-infected patients, pregnant women presented higher levels of MDA. These findings indicate that both HIV infection and pregnancy lead to lipid peroxidation, which is exacerbated when these two factors are combined. However, in non-pregnant women, the increased oxidative stress detected in HIV-positive women may be caused by the high levels of carbonylated proteins found, rather than MDA, as the later was not changed in the presence of infection.

In a study with healthy pregnant women [31], MDA concentrations were elevated in the first gestational trimester. In addition, Anderson et al. [36] demonstrated that high levels of MDA influenced the low birth weight of newborns from HIV-positive mothers compared to those without the infection.

Regarding the changes in SOD levels, infection influenced the activity of this enzyme in a greater way than pregnancy in all HIV-infected women. These women also presented higher CAT activity than the uninfected patients, but this activity was decreased during the gestational period. However, both infection and pregnancy factors seem to influence CAT activity.

For carbonylation, HIV-negative and non-pregnant HIV-positive women presented higher levels, what demonstrates that both factors are also related to higher protein oxidation and its modification. Moreover, both infection and pregnancy increased lipid peroxidation measured by MDA.

Finally, some limitations of our study should be addressed. Some confounding effects are present in this study, such as time of ART duration, time of HIV infection since diagnosis and oral nutrient supplementation that can modulate oxidative stress, as well as the participants' different lifestyles regarding physical activity and diet, for example, which have not been evaluated here. Besides, although increased oxidative stress is related to gestational complications such as preeclampsia and premature delivery, these outcomes are important to be studied but they were not investigated here.

Our study showed that HIV infection is responsible for increased SOD and CAT activities in both pregnant and non-pregnant women, increased MDA levels in pregnant, and changes in carbonylation in non-pregnant. As for pregnancy, although it did not cause changes in the activity of SOD and MDA, it led to increased CAT activity as gestational trimesters advanced, regardless of the infection status. Thus we conclude that HIV-infected pregnant women are under higher oxidative stress, however, the virus is not solely responsible for this process, in which there seems to be the involvement of other pregnancy-related mechanisms. More longitudinal study is needed in HIV pregnant women to bring out the clearer picture of redox imbalance in relation to HIV and pregnancy in the research environment.

## Supporting information

**S1 Data.**
(XLSX)

## Acknowledgments

We thank all the volunteers for participating in this study and the staff of the Specialized Outpatient Service for Infectious Diseases "Domingos Alves Meira" (SAEI-DAM), and Tropical Diseases Department and Laboratory, for their cooperation in the development of this study. We also thank Regina Moreto for helping with the laboratory tests, Dr José Eduardo Corrente for running the statistics analysis.

## Author Contributions

**Conceptualization:** Vanessa Martinez Manfio, Lenice do Rosário de Souza.

**Data curation:** Vanessa Martinez Manfio.

**Formal analysis:** Vanessa Martinez Manfio, Jessica Leite Garcia, Janaina de Oliveira Góis, Camila Renata Correa, Lenice do Rosário de Souza.

**Investigation:** Vanessa Martinez Manfio.

**Methodology:** Vanessa Martinez Manfio, Karen Ingrid Tasca, Lenice do Rosário de Souza.

**Project administration:** Vanessa Martinez Manfio, Lenice do Rosário de Souza.

**Resources:** Vanessa Martinez Manfio.

**Software:** Vanessa Martinez Manfio.

**Supervision:** Vanessa Martinez Manfio, Karen Ingrid Tasca, Camila Renata Correa, Lenice do Rosário de Souza.

**Validation:** Vanessa Martinez Manfio, Karen Ingrid Tasca, Lenice do Rosário de Souza.

**Visualization:** Vanessa Martinez Manfio, Camila Renata Correa.

**Writing – original draft:** Vanessa Martinez Manfio, Karen Ingrid Tasca.

**Writing – review & editing:** Vanessa Martinez Manfio, Lenice do Rosário de Souza.

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
