## [Decision Letter · Decision Letter 0]

1 Jul 2020

PONE-D-20-03191

“Pregnancy factor versus infection factor in HIV-infection and the redox balance”

PLOS ONE

Dear Dr. Martinez Manfio,

Thank you for submitting your manuscript to PLOS ONE. After careful consideration, we feel that it has merit but does not fully meet PLOS ONE’s publication criteria as it currently stands. Therefore, we invite you to submit a revised version of the manuscript that addresses the points raised during the review process.

The reviewers raised several concerns about the methodology of the study, including the sample size and statistical approach. They also requested a number of clarifications about the data set and some of the reported values/parameters. The reviewers' full comments can be viewed below.

We look forward to receiving your revised manuscript.

Kind regards,

Natasha McDonald, PhD

Associate Editor

PLOS ONE

Journal Requirements:

Additional Editor Comments (if provided):

Reviewers' comments:

Reviewer's Responses to Questions

**Comments to the Author**

1. Is the manuscript technically sound, and do the data support the conclusions?

Reviewer #1: Partly

Reviewer #2: Yes

2. Has the statistical analysis been performed appropriately and rigorously? 

Reviewer #1: No

Reviewer #2: Yes

3. Have the authors made all data underlying the findings in their manuscript fully available?

Reviewer #1: Yes

Reviewer #2: Yes

4. Is the manuscript presented in an intelligible fashion and written in standard English?

Reviewer #1: Yes

Reviewer #2: Yes

5. Review Comments to the Author

Reviewer #1: In the present article the authors tried evaluate pregnancy factor versus infection factor in HIV-infection and the redox balance. I have few observations and recommendations, those are as follows,

1- One of the major shortcoming of the manuscript is sample size to draw meaningful conclusion. The authors should include more cases and controls to substantiate their findings.

2- The main parameter of oxidative stress is total antioxidant capacity (TAC); the value of this parameter is necessary in this manuscript for evaluating antioxidant status.

3- Between groups Comparisons was presented in an ambiguous manner for example, the following comparison was made

Line 213 & 221: G1 (red): HIV-positive women, G2 (blue): HIV-negative women

4- The figures does not well drawn.

5- There are some mistakes in the manuscript for example:

Line 231: non-pregnant pregnant women at

Reviewer #2: The paper presents a protocol investigating the possible influence of both infection and pregnant factor in redox balance, an accrual and very important topic. The manuscript format is almost according to the authors’ instruction of the journal. The reported evidences related to the topic of the article are correct treated and used. The different aspects considered in the text follow a logical order. Nevertheless, there are some aspects of the paper which are not clear and others must be changed. Taking into account the previous research presented, inclusion and exclusion criteria, ethical issues and methods used are

correctly. Some grammatical and structure errors are found that must be corrected via helps of fluent lecture. Conflict of Interest and ethical issues are declared.

Page 6 line 108 Based on what this data was obtained. Clarify please. Other comorbidities could also influence on redox indexes as infection: hepatitis C virus infection, tuberculosis, etc or degenerative: hypertension, toxic habits. Did you consider during selection process?

Page 7 line 163 Some aspects need to be clarified in relation to the number of recruited patients and statistical method. Through redox indexes analytical process of validation some variability values are obtained in terms of precision and specificity. Those alpha and beta indexes are necessary to estimate the number of individual to obtain significant statistics with 95% of confidentiality. Accordingly to the indexes used in the study is required at least 25 individuals.

6. PLOS authors have the option to publish the peer review history of their article (what does this mean?). If published, this will include your full peer review and any attached files.

Reviewer #1: No

Reviewer #2: No

---

## [Author Response · Author response to Decision Letter 0]

22 Sep 2020

Dear reviewers, thank you very much for the questions and suggestions. Your comments were entirely pertinent and greatly improved the paper. Please find below the answers and modifications determined by the points raised in your revision. All the alterations triggered your comments are in are highlighted in yellow. 

Note: We suggest changing the title, according to a letter sent to the editor, for the purpose of better understanding by readers. “Redox imbalance is related to HIV and pregnancy”, rather than “Pregnancy factor versus infection factor in HIV-infection and the redox balance”. However, for this change, we await the approval of you and the editor.

Reviewer #1: 

In the present article the authors tried evaluate pregnancy factor versus infection factor in HIV-infection and the redox balance. I have few observations and recommendations, those are as follows.

1- One of the major shortcoming of the manuscript is sample size to draw meaningful conclusion. The authors should include more cases and controls to substantiate their findings.

The authors agree with the reviewer about the small sample number and that the inclusion of new patients could result in data that are more substantial. However, the inclusion was carried out according to the universe of eligible patients who were attended at our Health Service. Although it serves about 700 HIV-individuals, the number of pregnant women is restricted, as well as their interest in participating in the study. Part of this information was included in the methodology section (lines 99-103). Additionally, the study ended last year, since it was a doctoral thesis that was completed. Thus, unfortunately, it is not possible now to increase the sample size.

2- The main parameter of oxidative stress is total antioxidant capacity (TAC); the value of this parameter is necessary in this manuscript for evaluating antioxidant status.

The total antioxidant capacity (TAC) is an analyte frequently used to assess the antioxidant status of biological samples and can evaluate the antioxidant response against the free radicals produced in a given disease (Spectrophotometric assays for total antioxidant capacity (TAC) in dog serum: an update doi: 10.1186 / s12917-016-0792-7), therefore not being the main parameter of oxidative stress. Furthermore, considering that pregnant women take vitamin supplements and may have a different diet, TAC could overestimate the baseline protection value, as it represents both endogenous and exogenous antioxidants. The aim of this paper was to evaluate the influence of both HIV and pregnancy, in the redox system it comprises in oxidation and antioxidant products. For this, we measure the oxidation products of lipids (malondialdehyde) and proteins (carbonylation). When these analyzes present high values, it is inferred that the reactive oxygen species are being produced in large quantities causing an oxidation of biomolecules which damage tissues and organs leading to the appearance of diseases. However, the other side of the redox system was also checked in this paper, the activity of antioxidant enzymes (SOD and CAT) that has the function of neutralizing reactive species, preventing the oxidation of biomolecules. The TAC data does not reflect damage to biomolecules, so it was not necessary to measure in this paper, we believe that the parameters evaluated respond well to the proposed objective.

3- Between groups Comparisons was presented in an ambiguous manner for example, the following comparison was made Line 213 & 221: G1 (red): HIV-positive women, G2 (blue): HIV-negative women

The comparison made on line 213 refers to the activity of antioxidant enzymes (SOD and Catalase) between G1 (red): HIV-positive women, and G2 (blue): HIV-negative women´. The comparison made on line 221 refers to oxidation markers (malondyaldeide and carbonilation). To better clarify, the text was corrected in the paper (lines 229-232 and 238-241).

4- The figures does not well drawn.

We did readjustments in the figures 1 and 2, according to your suggestion.

5- There are some mistakes in the manuscript for example: Line 231: non-pregnant pregnant women at.

The sentence has been corrected (lines 251-252).

Reviewer #2: 

The paper presents a protocol investigating the possible influence of both infection and pregnant factor in redox balance, an accrual and very important topic. The manuscript format is almost according to the authors’ instruction of the journal. The reported evidences related to the topic of the article are correct treated and used. The different aspects considered in the text follow a logical order. Nevertheless, there are some aspects of the paper which are not clear and others must be changed. Taking into account the previous research presented, inclusion and exclusion criteria, ethical issues and methods used are correctly. Some grammatical and structure errors are found that must be corrected via helps of fluent lecture. Conflict of Interest and ethical issues are declared.

1- Page 6 line 108 Based on what this data was obtained. Clarify please. Other comorbidities could also influence on redox indexes as infection: hepatitis C virus infection, tuberculosis, etc or degenerative: hypertension, toxic habits. Did you consider during selection process?

None of these women had co-infections or opportunistic diseases and two of them were smokers in the moments analysed, which did not influence our results. This information was inserted in paper (lines 184-185).

2- Page 7 line 163 Some aspects need to be clarified in relation to the number of recruited patients and statistical method. Through redox indexes analytical process of validation some variability values are obtained in terms of precision and specificity. Those alpha and beta indexes are necessary to estimate the number of individual to obtain significant statistics with 95% of confidentiality. Accordingly to the indexes used in the study is required at least 25 individuals.

The inclusion was carried out according to the universe of eligible patients who were attended at our Health Service. Although it serves about 700 HIV-individuals, the number of pregnant women is restricted, as well as their interest in participating in the study. Therefore, the study did not have a sample calculation, as all pregnant women belonging to the Service were invited to participate in the study. Part of this information was included in the methodology section (lines 99-103).

---

## [Decision Letter · Decision Letter 1]

5 Feb 2021

PONE-D-20-03191R1

Redox imbalance is related to HIV and pregnancy

PLOS ONE

Dear Dr. Martinez Manfio,

Thank you for submitting your manuscript to PLOS ONE. After careful consideration, we feel that it has merit but does not fully meet PLOS ONE’s publication criteria as it currently stands. Therefore, we invite you to submit a revised version of the manuscript that addresses the points raised during the review process.

The manuscript has been re-evaluated by three reviewers, and their comments are available below. You will see that the reviewers considered your response to previous comments thorough. However, the reviewers have also raised concerns and the manuscript will need revision before it can be considered for publication. I have outlined the key concerns noted by the reviewers below, but you should respond to all concerns mentioned by the reviewers in your response-to-reviewers document. 

The key concerns noted by the reviewers relate to the reporting of the methods and the interpretation of the findings. Specifically, Reviewer 4 raise concerns about the presence of comorbidities in the study population that would limit the interpretation of the relationship between HIV and pregnancy. Additionally, Reviewer 4 noted that the sample size calculation is not presented.

We look forward to receiving your revised manuscript.

Kind regards,

Danielle Poole

Academic Editor

PLOS ONE

Reviewers' comments:

Reviewer's Responses to Questions

**Comments to the Author**

1. If the authors have adequately addressed your comments raised in a previous round of review and you feel that this manuscript is now acceptable for publication, you may indicate that here to bypass the “Comments to the Author” section, enter your conflict of interest statement in the “Confidential to Editor” section, and submit your "Accept" recommendation.

Reviewer #1: All comments have been addressed

Reviewer #3: All comments have been addressed

Reviewer #4: All comments have been addressed

2. Is the manuscript technically sound, and do the data support the conclusions?

Reviewer #1: Yes

Reviewer #3: Partly

Reviewer #4: No

3. Has the statistical analysis been performed appropriately and rigorously? 

Reviewer #1: Yes

Reviewer #3: Yes

Reviewer #4: No

4. Have the authors made all data underlying the findings in their manuscript fully available?

Reviewer #1: Yes

Reviewer #3: Yes

Reviewer #4: No

5. Is the manuscript presented in an intelligible fashion and written in standard English?

Reviewer #1: Yes

Reviewer #3: Yes

Reviewer #4: No

6. Review Comments to the Author

Reviewer #1: (No Response)

Reviewer #3: The authors have tried to address some ambiguity found in the article by the previous reviewers.

The reviewers’ comments has been strictly revised and effected by the authors to meet with the journal criteria.

Also the title has been modified to suit the purpose of the findings.

I therefore recommend that the article be considered for publication in PLOS one journal.

Reviewer #4: I would like to congratulate the authors for the idea of the article that presents a relevant theme and of interest to the maternal and child area. However, I have doubts about the inclusion and exclusion criteria considered in the study. As described by the authors, the entire study sample came from an infectology unit. Thus, it seems to me that the presence of other comorbidities, infections and / or situations (such as tobacco use) (a fact already reported by the authors) presented by some individuals were not considered at the time of screening. This becomes worrying because it is already well discussed in the literature that most existing diseases and smoking itself lead to increased oxidative stress. Thus, there is no guarantee that the results found are specific to HIV and pregnancy. The ideal for the data to be indisputable is that the controls present the same characteristics as the cases, excluding only the object (s) studied and this needs to be well described in the study in the method and results sessions. Additionally, as there is no description of the sample calculation, there is no guarantee that the sample studied is sufficient to consider the extrapolation of the results presented.

7. PLOS authors have the option to publish the peer review history of their article (what does this mean?). If published, this will include your full peer review and any attached files.

Reviewer #1: No

Reviewer #3: **Yes: **Nkiruka Rose Ukibe

Reviewer #4: No

---

## [Author Response · Author response to Decision Letter 1]

12 Feb 2021

Review and Author Comments

Dear reviewers, thank you very much for the questions and suggestions. Your comments were

entirely pertinent and greatly improved the paper. Please find below the answers and modifications

determined by the points raised in your revision. All the alterations triggered your comments are

highlighted in blue in the new version of manuscript.

Editor Comment:

The key concerns noted by the reviewers relate to the reporting of the methods and the

interpretation of the findings. Specifically, Reviewer 4 raise concerns about the presence of

comorbidities in the study population that would limit the interpretation of the relationship

between HIV and pregnancy. Additionally, Reviewer 4 noted that the sample size calculation is

not presented.

Reviewer #4:

I would like to congratulate the authors for the idea of the article that presents a relevant

theme and of interest to the maternal and child area. However, I have doubts about the

inclusion and exclusion criteria considered in the study.

1- As described by the authors, the entire study sample came from an infectology unit. Thus,

it seems to me that the presence of other comorbidities, infections and / or situations (such as

tobacco use) (a fact already reported by the authors) presented by some individuals were not

considered at the time of screening. This becomes worrying because it is already well discussed

in the literature that most existing diseases and smoking itself lead to increased oxidative

stress. Thus, there is no guarantee that the results found are specific to HIV and pregnancy.

Our infectious diseases service (SAEI-DAM) specializes in HIV/AIDS, hepatitis and

HTLV. None of the included women had co-infections or opportunistic diseases.

Regarding other comorbidities and drugs use, only one patient had diabetes and other two

were smokers, which did not influence our results. We added all these information in the

results section, lines 199-202).

2- The ideal for the data to be indisputable is that the controls present the same characteristics

as the cases, excluding only the object (s) studied and this needs to be well described in the

study in the method and results sessions.

The authors agree with the reviewer about the need for homogeneity between the groups

so that the differences found are strictly attributed to the factors “HIV infection” and

“pregnancy”. So, in addition to the sociodemographic table already presented, we have

included the p values and another paragraph containing a more clearly information about

it (results section, lines 177-194). We noticed that only 3 parameters were not

homogeneous between the groups: age (however, we believe that the oxidative stress

markers would not undergo major changes, considering that all women were very young);

time of infection and time on ART (after all, most women were diagnosed with HIV in

prenatal care). The heterogeneity of these two variables was included in the study's

limitations (lines 369-370).

3- Additionally, as there is no description of the sample calculation, there is no guarantee that

the sample studied is sufficient to consider the extrapolation of the results presented.

We understand your concern about the lack of sample size calculation, however, as

previously mentioned, the inclusion was carried out according to the universe of eligible

patients who were attended at our Health Service. Therefore, according to our patient flow,

we made every effort to invite all pregnant women to this study, after careful analysis of

the inclusion and exclusion criteria.

Although it serves about 700 HIV-individuals, the number of pregnant women is restricted

(and can be seasonal), as well as their interest in participating in the study (a large number

of women refuse to participate in the study due to the 3 blood collections proposed in the

study design).

Perhaps an initial sample calculation could even show the need to include more participants

in this study; however, it would not be feasible with our reality to increase this number. In

addition, despite the small sample size, we were still able to demonstrate statistically

significant differences in relation to the various markers analyzed, which allowed us to

discuss the importance of considering both factors (pregnancy and HIV infection) in the

inflammatory / oxidative state of these women.

Additionally, the study ended in 2019, not allowing its continuation – of new inclusions

and evaluations - since it was a doctoral thesis that has already completed in the same year.

Thus, unfortunately, it is not possible now to increase the sample size.

---

## [Decision Letter · Decision Letter 2]

24 Mar 2021

PONE-D-20-03191R2

Redox imbalance is related to HIV and pregnancy

PLOS ONE

Dear Dr. Martinez Manfio,

Thank you for submitting your manuscript to PLOS ONE. After careful consideration, we feel that it has merit but does not fully meet PLOS ONE’s publication criteria as it currently stands. Therefore, we invite you to submit a revised version of the manuscript that addresses the points raised during the review process.

We look forward to receiving your revised manuscript.

Kind regards,

John S Lambert

Academic Editor

PLOS ONE

Journal Requirements:

Additional Editor Comments (if provided):

will consider acceptance once revisions made

Reviewers' comments:

Reviewer's Responses to Questions

**Comments to the Author**

1. If the authors have adequately addressed your comments raised in a previous round of review and you feel that this manuscript is now acceptable for publication, you may indicate that here to bypass the “Comments to the Author” section, enter your conflict of interest statement in the “Confidential to Editor” section, and submit your "Accept" recommendation.

Reviewer #3: All comments have been addressed

Reviewer #4: All comments have been addressed

2. Is the manuscript technically sound, and do the data support the conclusions?

Reviewer #3: Yes

Reviewer #4: Partly

3. Has the statistical analysis been performed appropriately and rigorously? 

Reviewer #3: Yes

Reviewer #4: Yes

4. Have the authors made all data underlying the findings in their manuscript fully available?

Reviewer #3: Yes

Reviewer #4: (No Response)

5. Is the manuscript presented in an intelligible fashion and written in standard English?

Reviewer #3: Yes

Reviewer #4: Yes

6. Review Comments to the Author

Reviewer #3: Reviewers comment to manuscript PONE-D-20-03191R2, entitled "Redox imbalance is related to HIV and pregnancy".

The authors presented the results of original research and met all other criteria to be eligible for publication in PLOS ONE except in the area of choosing the sample size population. However, I do suggest that the final decision of this manuscript is dependent on the Editor since this has been stressed severally by all the reviewers.

The acceptance of this article for publication in PLOS ONE is also left for the editor since the authors in every aspect did some justification in correcting and revising the manuscript according to the entire reviewers’ comments.

I also suggest that the authors explanations to the major issue in the manuscript which is the sample size population should be considered and accepted because, there is no way the authors could be able to go back to the field for recruitment of more new pregnant women to join to the size already had since 2019.

Permit me to stress that with my previous experiences in sample collection with HIV individuals, it is ‘VERY’ difficult to get the HIV individuals in full compliance for the three sample collections because of both environmental and social problems.

I must commend the authors for completely presenting the original results of this manuscript otherwise; they would have extrapolated and manipulated the sample size knowing full well that it would pose a serious handicap in the acceptance of the article.

I thereby suggest that the authors must recommend in their conclusion that more longitudinal work is needed in HIV pregnant women to bring out the clearer picture of Redox imbalance in relation to HIV and Pregnancy in the research environment

Reviewer #4: The study has some limitations that were readily discussed by the authors in the article. Thus, I consider it suitable for publication after the adjustments have been made.

7. PLOS authors have the option to publish the peer review history of their article (what does this mean?). If published, this will include your full peer review and any attached files.

Reviewer #3: No

Reviewer #4: No

---

## [Author Response · Author response to Decision Letter 2]

27 Mar 2021

PONE-D-20-03191, Review Report

MANUSCRIPT: “Redox imbalance is related to

HIV and pregnancy”

Botucatu, March 24th 2021.

PLOS ONE

---

## [Editor Report · Decision Letter 3]

30 Apr 2021

Redox imbalance is related to HIV and pregnancy

PONE-D-20-03191R3

Dear Dr. Martinez Manfio,

We’re pleased to inform you that your manuscript has been judged scientifically suitable for publication and will be formally accepted for publication once it meets all outstanding technical requirements.

Kind regards,

John S Lambert

Academic Editor

PLOS ONE

Additional Editor Comments (optional):

most corrections have been made, and acceptable for publication
---

## [Editor Report · Acceptance letter]

7 May 2021

PONE-D-20-03191R3 

Redox imbalance is related to HIV and pregnancy 

Dear Dr. Martinez Manfio:

I'm pleased to inform you that your manuscript has been deemed suitable for publication in PLOS ONE. Congratulations! Your manuscript is now with our production department. 

Kind regards, 

on behalf of

Dr. John S Lambert 

Academic Editor

PLOS ONE